# Measurement of Postoperative Quality of Pain in Abdominoplasty Patients—An Outcome Oriented Prospective Study

**DOI:** 10.3390/jcm12051745

**Published:** 2023-02-22

**Authors:** Sascha Wellenbrock, Matthias Michael Aitzetmüller, Marie-Luise Klietz, Philipp Wiebringhaus, Gabriel Djedovic, Tobias Hirsch, Ulrich M. Rieger

**Affiliations:** 1Department of Plastic and Reconstructive Surgery, Institute for Musculoskeletal Medicine, Westfälische Wilhelms-University, 48149 Münster, Germany; 2Department of Plastic Surgery, Department for Traumatology and Hand Surgery, University Hospital Münster, Albert-Schweitzer-Campus 1, 48149 Münster, Germany; 3Department of Plastic Surgery, Community Hospital Feldkirch, Carinagasse 47, 6800 Feldkirch, Austria; 4Department of Plastic Surgery, Agaplesion Markus Hospital, Wilhelm Epstein Str. 4, 60431 Frankfurt am Main, Germany

**Keywords:** body contouring, postoperative management, pain, abdominoplasty

## Abstract

(1) Background: Postoperative pain is a frequently underestimated complication significantly influencing surgical outcome and patient satisfaction. While abdominoplasty is one of the most commonly performed plastic surgery procedures, studies investigating postoperative pain are limited in current literature. (2) Methods: In this prospective study, 55 subjects who underwent horizontal abdominoplasty were included. Pain assessment was performed by using the standardized questionnaire of the Benchmark Quality Assurance in Postoperative Pain Management (QUIPS). Surgical, process and outcome parameters were then used for subgroup analysis. (3) Results: We found a significantly decreased minimal pain level in patients with high resection weight compared to the low resection weight group (*p* = 0.01 *). Additionally, Spearman correlation shows significant negative correlation between resection weight and the parameter “Minimal pain since surgery” (rs = −0.332; *p* = 0.013). Furthermore, average mood is impaired in the low weight resection group, indicating a statistical tendency (*p* = 0.06 and a Χ^2^ = 3.56). We found statistically significant higher maximum reported pain scores (rs = 0.271; *p* = 0.045) in elderly patients. Patients with shorter surgery showed a statistically significant (Χ^2^ = 4.61, *p* = 0.03) increased claim for painkillers. Moreover, “mood impairment after surgery” shows a dramatic trend to be enhanced in the group with shorter OP duration (Χ^2^ = 3.56, *p* = 0.06). (4) Conclusions: While QUIPS has proven to be a useful tool for the evaluation of postoperative pain therapy after abdominoplasty, only continuous re-evaluation of pain therapy is a prerequisite for constant improvement of postoperative pain management and may be the first approach to develop a procedure-specific pain guideline for abdominoplasty. Despite a high satisfaction score, we detected a subpopulation with inadequate pain management in elderly patients, patients with low resection weight and a short duration of surgery.

## 1. Introduction

Being essential for postoperative complications, morbidity, mortality as well as rehabilitation capacity, guideline-based pain therapy has become an integral part for almost all surgical disciplines [1]. 

While not only perioperative morbidity was found to be reduced by adequate pain medication, several studies describe a significant decrease in complications with a verifiable reduction of hospitalization days [1,2]. 

Therefore, postoperative pain management is essential not only for individual patients but chronification of underestimated postoperative pain represents an economic burden, with enormous potential for optimization.

Pain management can be divided into non-medicinal and medicinal factors.

While non-medicinal factors include psychological and physical procedures, such as the application of cold to reduce the swelling of an extremity after postoperative decongestion of an extremity after surgery, medical factors mainly focus on systemic pharmacotherapy.

Among these, based on international guidelines, treatment of severe to moderate pain should be based on a combination of opioids (tramadol, piritramide) and non-opioid analgesics (paracetamol, metamizole, NSAIDs, COX-2 inhibitors) [3].

Although this seems to be standardized, many studies have shown insufficient pain management [4,5]. For further standardized assessment and improvement, the “Quality Improvement in Postoperative Pain Therapy” (abbreviated as follows: “QUIPS”) as an interdisciplinary project was initiated. Being the world’s largest acute pain registry and including data on process and quality outcomes, this system allows collection, evaluation and improvement of acute pain therapy in participating institutions [6].

Although it is undoubted, that adequate pain management is essential for individual outcome, there exist almost no studies evaluating postoperative pain in plastic surgery. Especially in semi-elective surgeries, such as body contouring surgeries, characterized by large wound areas, postoperative well-being can lead to faster mobilization and reduction of hospitalization time. Nevertheless, hardly any literature is available on this topic.

Single case reports describe the existence of neuropathic pain syndromes of the N. iliohypogastricus and cutaneous femoris lateralis after abdominoplasty and their avoidability [7]. Feng et al. describe the reduction of pain by combination of local nerve blocks during abdominoplasty surgery [8].

Regarding pain medication, a recommendation of reduced opioid consumption after abdominal wall surgery can be found, but there exist no concrete guidelines and this recommendation has not been evaluated [9].

To sum up, although postoperative pain management has been excessively described to be of utmost importance for outcome and well-being, no standardized study within body contouring patients has been carried out up to now. 

Therefore, we used QUIPS in abdominoplasty patients for analyzing pain characteristics as well as to define risk factors for enhanced postoperative pain. 

## 2. Materials and Methods

This study was carried out following the guidelines of the declaration of Helsinki as well as by the dean of the university. 

### 2.1. Inclusion and Exclusion Criteria

All patients undergoing abdominoplasty according to Pitanguy at the Department for Plastic and Aesthetic Surgery, Reconstructive and Hand Surgery at the Markus Hospital in Frankfurt am Main from January 2010 to December 2015 were included in this study. 

For the study, all patients were excluded who underwent combination or revision procedures such as autologous breast reconstruction or repair of rectus diastasis. 

### 2.2. Data Collection

A standardized pain questionnaire (QUIPS) was carried out on postoperative day one by a single study nurse focusing on outcome (Appendix A) and processing parameters (Appendix B) using a Numeric Rating Scale from 0 to 10, or a dichotomous Yes/No categorization. This questionnaire was filled out manually under surveillance. Additionally, preoperative anesthesiologic assessments were screened for general data such as age, sex, weight and for specific risk factors, comorbidities and ASA score. Surgical protocols were screened for resection weight as well as for surgery time. 

### 2.3. Surgical Procedure

All surgical procedures were carried out by one senior doctor with one or two residents. The surgical procedure was standardized to prevent any technical related bias: Preoperatively, the patient is marked in a standing position to define the resection lines.After proper positioning, the surgical area is sterilely covered.The skin incision is made with the scalpel and the subcutaneous preparation by using the monopolar diathermy.The belly button is incised and sutured cranially with silk as holding suture.Epifascial dissection of the fat-skin soft tissues below and above the umbilicus up to the xiphoid while sparing the lateral sub- and intercostal perforator vessels. Hereby no focus is given on nerve sparing.Insertion of wound drains and drainage at the mons pubis.Collapsing the patient at the hip and re-defining the resection area.Resection of the skin fat flap and adaptation of the skin and tissue with Vicryl 2-0 and 3-0.Placement of the new umbilical position and suturing of the umbilicus with Vicryl 4-0 subcutaneous and Prolene 4-0.Wound closure with continuous intradermal Biosyn 3-0 suture and sterile wound dressing, abdominal belt.

### 2.4. Pain Mangement

All patients received pain medication via a standardized protocol following the official German guidelines for pain management (Appendix C) [3].

### 2.5. Statistical Analysis

Data analysis was performed using SPSS version 22.0 (IBM Corporation, New York, NY, USA). Initially, the database created by QUIPS was completed with the operation- and patient-related data. After conversion, analysis of descriptive data was performed. All data are given as mean and standard deviation (=SD). Nominal-distributed data were analyzed using Pearson’s chi-square test, Fisher’s exact test, and Spearman-rho correlation.

To analyze variables and individual subgroups of this population, a univariate and multivariate correlation analysis as well as the Mann Whitney-U test and the Kruskal Wallis test were used. The median was used for division of groups. A *p* level of <0.05 was considered as statistically significant. 

## 3. Results

### 3.1. Demography

In total, 268 patients underwent abdominoplasty within the given timeframe. Further, 110 were excluded due to surgical technique. Of the remaining 158 patients, 55 showed a complete QUIPS and gave written consent for participation. Among those, 41 (75%) were female and 14 (25%) were male, aged between 21 and 67 years. Mean age and mean height was 42.93 ± 9.9 and 169.22 cm ± 8.13 cm, retrospectively. Average weight of patients was 87.05 kg ± 19.73 kg. 

### 3.2. Surgical Procedures

Average resection weight was 2913 g ± 2226 g and average duration of surgery was 129.49 ± 37.48 min, with 215 min representing the longest and 56 min the minimal surgical time. 

### 3.3. Preoperative Measurements

Patients were preoperatively classified using the ASA sore. Thereby, five (9%) subjects were categorized as ASA-I, 42 (76%) as ASA-II, and eight (15%) as ASA-III. No ASA-IV or V subjects underwent surgery. In total, seven (13%) patients stated regular intake of painkillers before surgery, due to chronic diseases.

### 3.4. QUIPS Outcome Parameters Overall

Table 1 depicts the overall outcome results in our population.

Mean pain on exertion was reported as 4.42 ± 1.54, with the maximum pain being 5.35 ± 2.04 and the minimum pain 1.95 ± 1.43. Thirty-eight of the 55 patients (69%) exceeded the pain level of 4, which normally is considered as the tolerance pain threshold for demanding pain killers. Patient satisfaction was reported to be 11.95 ± 3.03 in average.

In terms of mobility, 34 (62%) patients reported being significantly limited due to pain. When breathing or coughing, 27 (49%) mentioned pain while they in- or exhaled. Thirteen of the respondents (24%) felt their sleep was disturbed and 12 (23%) reported their mood being affected by postoperative pain. Twenty-six (47%) of all respondents reported postoperative pain fatigue. Both nausea and vomiting were mentioned by only 12 (23%) and nine (13%), respectively. Chronic pain was previously described by seven (13%) of the total collective.

Nevertheless, only 13 (24%) demanded extra painkillers. In addition, 26 (47%) of all respondents felt postoperative pain fatigue. 

### 3.5. QUIPS Process Parameters Overall

Preoperatively, midazolam 7.5 mg per os was offered to all patients as a sedative but was taken only by six patients (11%).

For intraoperative pain relief, an opioid (Sufentanil) was used in all except five (9%) cases. Non-opioids were used in seven patients (13%). 

In the postoperative care unit, opioids were used in 53 (96%) patients and in 35 (64%) patients, non-opioids were injected intravenously. 

Postoperatively, midazolam was injected in 35 (64%) cases. Diclofenac was used as a second-line agent in five cases (9%). In addition, 15 (27%) patients did not require any analgesics. In the majority of subjects (*n* = 31, 56%), no opioid was needed. However, in 24 patients (44%), Piritramid 7.5 mg i.v. was used postoperatively.

### 3.6. Influence of Surgical Parameters on Postoperative Pain Outcome Parameters

For a profound analysis of the quality of postoperative pain therapy, subpopulations were created based on surgical parameters and patient characteristics. Therefore, means were used as dividing factors: 

Resection weight: Mean weight is 2180 g,

Age: Mean age is 43 years,

Duration of surgery: Mean operation time 125 min.

#### 3.6.1. Resection Weight

When considering the factor “resection weight”, we found significant decreased minimal pain in patients with high resection weight compared to the low resection weight group (*p* = 0.01) as shown in Table 2. Additionally, Spearman correlation analysis shows a significant negative correlation between resection weight and the parameter “Minimal pain since the Surgery” (Spearman Rho coefficient rs = −0.332; significance value *p* = 0.013 *).

Additionally average mood was impaired in the low weight resection group. A *p*-value of 0.06 and a Χ^2^-value of 3.56 indicate a trend without being statistically significant.

#### 3.6.2. Age

With a range of 21 to 67 years, the average age is 42.93 ± 9.9 years with a median of 43 years.

We found statistically significant higher maximum reported pain scores (Spearman Rho coefficient rs = 0.271; significance value *p* = 0.045 *) in older patients, indicating enhanced pain within this group as shown in Table 3. 

#### 3.6.3. Duration of Surgery

The average operation time in this study was 129.49 ± 37.48 min, with a range of 159 min and a median of 125 min.

Table 4 shows that Patients with a shorter surgery had a statistically significant (Χ^2^ = 4.61, *p* = 0.03 *) increased claim for painkillers. Additionally, “mood impairment after surgery” showed a dramatic trend to be enhanced in the group with shorter OP duration (Χ^2^ = 3.56, *p* = 0.06). 

## 4. Discussion

Aimed at a redefinition of the body contour, abdominoplasty is performed by wide undermining of tissue of the abdominal wall with its high density of thoracolumbal sensitive nerves [10]. Due to this fact, it remains uncertain how much nerve injury is caused by wide preparation. Pogatzski-Zahn emphasize that intraoperative nerve irritation can cause chronic pain syndromes [11] and nerve injury during this procedure represents an underestimated problem [12].

Despite SOPs and guidelines, as well as an improved patient education, pain is constantly considered as the fifth vital sign [13] and its postoperative management is far from being sufficient [5,6].

With a few exceptions in plastic surgery, there exists no literature of postoperative pain management in standard procedures. One of the reasonable tools for pain relief, published with a small cohort, is additional regional nerve block of the area of interest (e.g., breast [14] or abdomen [15]) or the use of lidocaine-infusing pain pumps [16]). These steps lead to reduced hospital stay, overall pain reduction and a reduced pain medication compared to the control group [17].

Nevertheless, literature of an analysis of pain quality in abdominoplasty with its outcome parameters as a benchmarking tool for evaluation is missing.

Therefore, for the first time, we implemented QUIPS for the analysis of plastic surgery pain management. 

In our analysis, the mean maximal pain intensity overall was 5.35 out of 10 in a NRS. Pain levels in 38 subjects (69%) were above a value of 4 and according to the S3-guidelines of perioperative pain treatment [18], therefore needed to be addressed for prevention of long-term functional impairment.

By comparing our cohort result of maximal pain intensity with other different common procedures, such as an appendectomy 5.20 or a functional endoscopic sinusitis surgery 3.96, this level shows a relatively high maximum pain level. Nevertheless, in comparison to traumatological procedures (such as the cruciate ligament-plasty up to 6.0 out of 10), it shows lower maximum pain intensity [19]. 

We used the median to split participants into two subpopulations regarding their resection-weight, age and duration of surgery, being consistent with our clinically experience. 

The cohort with medical indication can be divided into patients with a relevant, functional impairing dermatochalasis of the abdomen, a longer operation time, a higher ASA status and generally the younger patient after bariatric surgery or self-induced massive weight loss.

In contrast to the previous mentioned, the aesthetic patient population comes along with a moderate low resection weight, less ptosis of skin, normally younger women after pregnancy asking for a mommy makeover or the “best ager”, who are older but healthier with a profound focus on their outward appearance and self-motivated initiative for a tummy tuck.

A direct comparison between those two groups is desired and needs further research.

Nevertheless, we evaluated the outcome parameters in relation to the patient specific parameters of our subgroups.

A higher resection weight correlates with a higher bodyweight and this is regarded as a predictor for a higher complication rate in abdominoplasty [20,21].

Interestingly, none of these studies analyzed the influence of the factor “resection weight” itself (in our study ranging from 610 g up to 9600 g).

In contrast to our prediction, our observations show that high resection weight goes along with significantly less pain markers and the mood impairment of patients with low resection weights.

There are no data in literature which confirm those findings for this procedure.

In reduction mammaplasty, Strong et al. found that patients with higher resection weights have significantly less pain than those with low resection rates [22].

A reasonable explanation of our results is the decreased sensibility of the abdominal wall in patients with a large apron of fat and high resection weight. We assume that patients with hanging bulge of skin and fat tissue have a higher basic pain level and they tolerate postoperative surgical pain better than in lower resection weights [23]. We can further postulate that a chronic local hypoxia, with a consecutive increasing lactate level and a lowering of the pH-value in the apron of fat, elevates the excitation threshold of the peripheral nervous system in this local bulge, causing delayed or even failing to trigger an action potential. This hypothesis has been initially drawn by Kim et al., postulating an ischemic-related pain mechanism when showing a significantly elevated lactate level in postoperative wounds [24].

However, one has to be aware that resection weight can be high in patients with massive weight loss with normal BMI. 

The correlation analysis shows that older patients have significantly higher maximal pain levels, which is contrary to current literature, emphasizing young age to be a risk factor for postoperative pain [11,25,26,27]. A higher tolerance of pain is awarded to older patients with reduced analgetic consumption, reduced nociceptive activity and lower needs for morphine medication [28]. Elderly patients vary in distribution of medication, metabolism and excretion of pain medication compared to young people [29].

Morphology-wise, older skin shows dermal atrophy with less perfusion, less elastic fibers and less Meissner and Vater Pacini bodies, resulting in a lower tactile and pressure sensibility [30,31]. These factors cause decreased tolerance of shear forces and tension taking place in an abdominoplasty. Young age as a risk factor for developing pain could not be confirmed in our study.

Enlarged duration of surgery is broadly accepted as a risk factor for pain as well [24,32]. Interestingly, patients with lower operation time have significantly higher desire for pain medication and a high tendency of mood disturbance. While those probands with lower operation time mostly have a lower resection weight, reflecting aesthetic abdominoplasty cases, the expectations for this procedure might be higher due to their payment and probably the conditions of their pain-ranking are much stricter than those in the insurance paid comparative group. Additionally, due to shorter surgery-time, subjects mobilize themselves earlier and this might lead to wound tension, with worse pain outcome portrayed by an increased claim for painkillers and mood disturbance [33]. The removal of the suction drain itself causes pain, discomfort and anxiety [34]. The survey took place on the first postoperative day. Patients with shorter duration of surgery are probably more likely to get rid of suction drains earlier in temporary connection to the QUIPS interview, causing high pain intensity.

The statement: “Longer operation time is a predictor for increased postoperative pain” can be rejected in our investigation.

Limitation is the relatively small cohort and a single center study, as well as a relatively small time frame, which needs to be verified in ongoing investigations. Additionally, due to the structure of QUIPS, no conclusion can be drawn about pre-existing chronic disorders influencing postoperative pain medication. 

## 5. Conclusions

Abdominoplasty is a standardized procedure that is, due to its large wound area, well suited for the evaluation of pain levels. QUIPS has proven to be a successful tool for the first elevation of postoperative pain quality in abdominoplasty procedures.

Despite a high overall satisfaction score, we detected a subpopulation with inadequate pain management in elderly patients, patients with low resection weight and a short duration of surgery. To what extent newly found risk factors will be deemed suitable to adapt tailored pain management requires further study to pave the way for a procedures-specific pain guideline.

## Figures and Tables

**Table 1 jcm-12-01745-t001:** Detailed QUIPS outcome parameters overall after abdominoplasty.

Variable	Mean	SD	
Pain on ambulation (0 to 10)	4.42	1.54	
Maximum pain intensity (0 to 10)	5.35	2.04	
Minimum pain intensity (0 to 10)	1.95	1.43	
Satisfaction with pain therapy (0 to 15)	11.95	3.03	
Variable	Value	N	%
Mobility impaired due to pain	No	21	38.18
	Yes	34	61.82
Breathing impaired due to pain	No	28	50.91
	Yes	27	49.09
Night awakening	No	42	76.36
	Yes	13	23.64
Mood Disturbance	No	43	78.18
	Yes	12	21.82
Desire for pain medication	No	42	76.36
	Yes	13	23.64
Drowsiness since surgery	No	29	52.73
	Yes	26	47.27
Nausea since surgery	No	43	78.18
	Yes	12	21.82
Vomiting since surgery	No	46	83.64
	Yes	9	16.36
Preoperative Pain management counselling	No	48	87.27
	Yes	7	12.73

**Table 2 jcm-12-01745-t002:** Relation between outcome parameters and low-/high resection weights.

	≤2180 g		>2180 g		
	Mean	SD	Mean	SD	*p*
Pain on ambulation	4.54	1.79	4.30	1.23	0.42
Maximum pain intensity	5.32	2.20	5.37	1.90	0.99
**Minimum pain intensity**	**2.43**	**1.29**	**1.44**	**1.42**	**0.01 ***
Satisfaction with pain therapy	11.75	3.16	12.15	2.94	0.67
		≤2180 g	>2180 g	χ^2^	*p*
Mobility impaired due to pain	No	9	12	0.88	0.41
	Yes	19	15		
Breathing impaired due to pain	No	13	15	0.46	0.50
	Yes	15	12		
Night awakening	No	20	22	0.77	0.38
	Yes	8	5		
**Mood Disturbance**	**No**	**19**	**24**	**3.56**	**0.06**
	**Yes**	**9**	**3**		
Desire for pain medication	No	21	21	0.06	0.81
	Yes	7	6		
Drowsiness since surgery	No	14	15	0.17	0.68
	Yes	14	12		
Nausea since surgery	No	22	21	0.01	0.94
	Yes	6	6		
Vomiting since surgery	No	24	22	0.18	0.73
	Yes	4	5		
Preoperative Pain Management counselling	No	24	24	0.12	0.52
	Yes	4	3		
**Spearman correlation analysis resection weight**					
Pain on ambulation		Spearman’s rho		−0.113	
		Significance level		0.413	
		N		55	
Maximum pain intensity		Spearman’s rho		−0.030	
		Significance level		0.828	
		N		55	
**Minimum pain intensity**		**Spearman’s rho**		**−0.332**	
		**Significance level**		**0.013 ***	
		**N**		**55**	
Satisfaction with pain therapy		Spearman’s rho		0.005	
		Significance level		0.973	
		N		55	

Bold is to express the significance or trend. * *p* < 0.05.

**Table 3 jcm-12-01745-t003:** Relation between outcome parameters and age-difference.

	≤43 Years		>43 Years		
	M	SD	M	SD	*p*
Pain on ambulation	4.33	1.63	4.52	1.45	0.57
Maximum pain intensity	4.97	2.19	5.80	1.78	0.13
Minimum pain intensity	2.10	1.52	1.76	1.33	0.40
Satisfaction with pain therapy	11.30	3.09	12.72	2.84	0.07
		≤43 Years	>43 Years	χ^2^	*p*
Mobility impaired due to pain	No	13	8	0.74	0.39
	Yes	17	17		
Breathing impaired due to pain	No	17	11	0.88	0.35
	Yes	13	14		
Night awakening	No	22	20	0.34	0.56
	Yes	8	5		
Mood Disturbance	No	21	22	2.59	0.11
	Yes	9	3		
Desire for pain medication	No	22	20	0.34	0.56
	Yes	8	5		
Drowsiness since surgery	No	13	16	2.34	0.13
	Yes	17	9		
Nausea since surgery	No	23	20	0.09	0.77
	Yes	7	5		
Vomiting since surgery	No	25	21	0.00	0.95
	Yes	5	4		
Preoperative Pain Management counselling	No	25	23	0.91	0.44
	Yes	5	2		

**Table 4 jcm-12-01745-t004:** Relation between outcome parameters and operation time.

	≤125 min		>125 min		
	Mean	SD	Mean	SD	*p*
Pain on ambulation	4.61	1.59	4.22	1.48	0.22
Maximum pain intensity	5.54	2.17	5.15	1.92	0.41
Minimum pain intensity	1.86	1.51	2.04	1.37	0.61
Satisfaction with pain therapy	11.54	3.28	12.37	2.75	0.30
		≤125 min	>125 min	χ^2^	*p*
Mobility impaired due to pain	No	13	8	1.64	0.20
	Yes	15	19		
Breathing impaired due to pain	No	16	12	0.89	0.35
	Yes	12	15		
Night awakening	No	20	22	0.77	0.53
	Yes	8	5		
Mood Disturbance	No	19	24	3.56	0.06
	**Yes**	**9**	**3**		
**Desire for pain medication**	**No**	**18**	**24**	**4.61**	**0.03 ***
	**Yes**	**10**	**3**		
Drowsiness since surgery	No	13	16	0.91	0.42
	Yes	15	11		
Nausea since surgery	No	22	21	0.01	1.00
	Yes	6	6		
Vomiting since surgery	No	23	23	0.09	1.00
	Yes	5	4		
Preoperative Pain Management counselling	No	26	22	1.60	0.25
	Yes	2	5		

Bold is to express the significance or trend. * *p* < 0.05.

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
