# Peer review of "Measurement of Postoperative Quality of Pain in Abdominoplasty Patients—An Outcome Oriented Prospective Study"

_jcm, 2023, doi:10.3390/jcm12051745_

Round 1

Reviewer 1 Report

Measurement of postoperative quality of pain in abdominoplasty patients - An outcome oriented prospective study

Wellenbrock S, Aitzetmüller M, Klietz M-L, Wiebringhaus P, Djedovic G, Hirsch T  and Rieger U

JCM

 In general this is an interesting prospective cohort report addressing postoperative quality of pain in abdominoplasty patients according to QUIPS. It reveals a subpopulations with worse pain scores.

Some essential information with respect to methodology and contextual circumstances are however not decribed. Furthermore grammar is untidy and reference numbers are not according to author instructions. Language should be checked.

 Abstract

Clearly written. One comment: Lower weight resection is correlated to impared mood. And patients with shorter surgery showed a statistically significant (Χ2 =4.61, p=0.03) increased claim for painkillers. It seems to me that these results are unexpected and authors suggest that pain management was bad in these patients. This last part of the conclusion is imaginable but an explanation that may lead to recommendation of better pain management.

Introduction

L38 Being essential for postoperative complications, morbidity mortality as well as rehabilitation capacity,…

Here lacks a comma between “morbidity mortality”.

L43-44 Therefore, postoperative pain management is essential not only for individual patients but also represents an economic burden,…

I do understand what authors intend to tell us, but postoperative pain management itself does not represent an economic burden. Please rewrite.

L47-48 While non-medicinal factors include psychological and physical procedures, such as the application of cold to reduce the swelling of an extremity after postoperative decongestion of an extremity after surgery.

This sentence seems not to be finished.

L67  Feng et al. describes the reduction of pain

Decribes does not need the letter s

Data collection

What is the primary outcome and what are the secondary outcomes? I think not all outcome criteria and their measures or units are addressed here, such as patient satisfaction.

L103-104 5. Epifascial dissection of the fat-skin soft tissues below and above the umbilicus up to the xiphoid while sparing the lateral sub- and intercostal perforator vessels

Did surgeons also try to spare the accompagning intercostal nerves?

The surgical procedure is described, but pain management not. This should be described in perspective of this study and also if there are various regimes or not.

At what moments are patients asked to assess their pain and in what way did they? On paper, digitally, otherwise? This should be desrcibed.

Results

L139-140 In total 7 (13%) of patients stated regularly intake of painkillers before surgery, due to chronic diseases.

Can authors describe if there were patients with chronic pain disorders? If so, which and what treatment did they have before starting the surgery?

L141-143 Mean pain on exertion is reported as 4.42 ± 1.54 with the maximum pain being 5.35 ±  2.04 and the minimum pain 1.95 ± 1.43. 38 of the 55 patients (69%) reach the tolerance pain threshold of 4/10 or exceed it.

This sentence does not run well. Why should 4 be a tolerance treshold? Usually the score on a numeric or verbal rating scale of 4 is considered to mean that pain is going from mild to moderate intensity. Please comment on this.

L143 Patient satisfaction is reported to be 11.95 ± 3.03 in average. 11.95?

On what scale is this?

L145 …while 28 patients (51%) denied this.

This suggests that patients deny having pain, while they probably do not report pain. Please comment on this.

L156 (Sufentayl): incorrectly spelled

L160 Postoperatively midazolam was injected in 35 (64%) cases.

Can authors explain why this was done? Why do so many patients need postoperatively sedative agents? What effect does this have on patient reporting, as postoperative pain is what the patients have to assess.?

Table 1

Satisfaction with pain therapy (0 to 15)

55

11,95

3,03

A scale of 0-15 is rather unusual. Can authors explain why such a scale is used? And why not 0-10 as in other cases? What effect does offering this alternative scale have on patients judgement and their scores?

I suggest the difference in variables first expressed in Mean and SD and thereafter expressed in N and %  is made more prominent. Or split in 2 tables/

Influence of surgical parameters on postoperative pain outcome parameters

L166-171 subpopulations were created based on surgical parameters and patient characteristics. Therefore means were used as dividing factors:

Resection weight: Mean weight is 2180 g

Age: Mean age is 43 years 170

Duration of surgery: Mean operation time 125 min

Why name them subpopulations? To my opinion they are not, but characteristics and variables and are presented in a dichotomous way by dividing them in two groups. Please comment on this.

Table 2, 3 and 4 are confusing as well. I suggest to split this in two tables or to make more obvious that these table consist of two parts.

L216 Pain levels in 38 probands (69%) were above a value of 4…

What does ‘probands’ mean?

L216 S3-Guidelines of perioperative pain treatment

S3-guideline is an unknown term, although a reference is provided. Can authors give a more appropriate description?

L219 On the one hand compared to an appendectomy 5.20….

Does this refer to pain scale (from 0-10)? Authors should make this clear.

Discussion

L219-222 On the one hand compared to an appendectomy 5.20 or functional endoscopic sinusitis surgery 3,96 this level in our cohort shows higher maximum pain level but in comparison to traumatological procedures (such as the cruciate ligament-plasty up to 6,0 out of 221 10) its shows lower maximum pain intensity19.

Please mention that nbumbers are pain scores?

L226-228 The Patient with medical indication with a relevant functional impairing the 226 dermatochalasis at the abdomen, with a longer operation time, the higher ASA status, and 227 generally the younger patient after bariatric surgery or self-induced massive weight loss.

This sentence seems not to be complete.

L229-232 The aesthetic patient population with a moderate low resection weight, less ptosis of 229 skin, normally younger women after pregnancy asking for a mommy makeover or the “best ager”, who are older but healthier with a profound focus on their outward appearance and self-motivated initiative for a tummy tuck.

Not complete also?

L238-239 Interestingly none of these study’s analysed the influence of the factor “resection weight” itself (in our study ranging from 610 g up to 9600 g).

To what studies do authors refer? And study’s should be studies?

L243 There are no data in literature, which confirm those findings for this procedure.

Please make explicit what “those findings” are.

L24-242 In contrast to our prediction our observations show that high resection weight goes 240 along with significant less pain markers and the mood impairment of patients with low 241 resection weights.

Should be: compared to patients with low resection weights?

L246-247 A reasonable explanation of our results is the decreased sensibility of the abdominal wall in patients with a large apron of fat and high resection weight.

Can authors refer to any literature that supports such a suggestion? Or next should connect directly?

L248 We assume that patients with hanging bulge of skin and fat tissue have a higher basic

Why starting a new alinea? This sentence is in alignment with the last sentence just before.

L254 a ischemic-related…

Should be an ischaemic-related…

L261 A higher tolerance of pain is awarded to older patients with a reduced analgetic…

Connect directly to last sentence before.

L261-263 A higher tolerance of pain is awarded to older patients with a reduced analgetic consumption, a reduced nociceptive activity and lower needs for morphine medication25. Elderly patients vary in distribution of medication, metabolism and excretion of pain medication compared to young people26.

There is ample literature that older patients experience more pain compared to younger patients. Please comment on this.

L2656-267 Morphology wise older skin shows dermal atrophy with less perfusion, less elastic fibres and less Meissner- and Vater Pacini bodies resulting in a lower tactile and pressure sensibility27,28.

Do authors mean: resulting in a lower tactile and pressure sensibility tresholds? Otherwise I do noit understand.

 L274-280 The expectations for this procedure due to their payment is higher and probably the conditions of their pain-ranking are much stricter than those in the insurance paid comparative group. Shorter operation time means decline of narcotic effect and less cumulative analgetic dose than longer duration of surgery. Due to less narcosis probands mobilise themselves earlier and this leads to wound tension with worse pain outcome portrayed by increased claim for painkillers and their mood disturbance. The removal of suction drain itself causes pain, discomfort and anxiety30.

Authors involve various factors in an hypothetic explanation theory that may be questioned. Please comment on this.

L184-185 The statement: “Longer operation time is a predictor for increased postoperative pain” can be rejected.

Authors should bring this only into the perspective of their study results.

L286 Limitation is the relatively small cohort and a single center study, which needs to be verified in ongoing investigations.

Good to address limitations, however, authors discuss only this limitation and I suggest them to be more critical and present and discuss more limitations,  that there are.

Conclusions

-

Author Response

Dear Reviewer 1,

Please  kindly see the attachment. All changes are highlighted in green

Reviewer 2 Report

In this prospective study, 55 subjects who underwent horizontal abdominoplasty were included. Pain assessment was performed by using the standardized questionnaire of the Benchmark Quality Assurance in Postoperative Pain Management (QUIPS).
Surgical, process and outcome parameters were then used for subgroup analysis.
Despite a high satisfaction score, the authors detected a subpopulation with inadequate pain management in elderly patients, patients with low resection weight and a short duration of surgery.

This is a well-designed study and of relevance for the clinicians. Some comments:

1. Why rectus diastasis repair was not included in this study?

2. How about postoperative compression garment and lymph drainage? What was the postoperative protocol?

3. Would the authors consider ultrasound therapy/radiofrequency after the treatment?

4. Would hyperbaric oxygen therapy after surgery improve pain sensation/outcome?

Author Response

Dear Reviewer 2,

Please  kindly see the attachment. All replies are highlighted in green.

Round 2

Reviewer 1 Report

I thank the authors for their reply. Many issues are addressed, but to my opinion in general the discussion remains of quite poor quality with unsupported suggestions. 

L139-140 In total 7 (13%) of patients stated regularly intake of painkillers before surgery, due to chronic diseases.

Can authors describe if there were patients with chronic pain disorders? If so, which and what treatment did they have before starting the surgery?

We thank the reviewer for the important comment. 13% of patients marked the question E14 of the QUIPS as you may see in the Appendix B “Have you ever had chronic pain before the operation” They only mentioned this but there was no hint for further/extensive medication treatment in this group.

It is a pity that Quips does not provide further information about pre-existing chronic disorders that ask for painkillers, as this might contribute to understanding the postoperative course in these patients.

 ------------------------------

 L143 Patient satisfaction is reported to be 11.95 ± 3.03 in average. 11.95?

On what scale is this?

There is the Question E13 in the QUIPS which has a patient satisfaction scale from 0-15

Why is this a scale  from 0-15, while usually these scales run form 0-10?

 -------------------------------

S3-guideline is an unknown term, although a reference is provided. Can authors give a more appropriate description?

We thank the reviewer for this important comment. The german medical guideline system contents of 3 levels of evidence. For this reason this S3-Guideline is regarded as a standard in perioperative pain treatment.

I thank the authors for this explanation. However, for the international reader S3 is still unknown. So authors should give a reference or a short explanation in the text.

 ------------------------------

 The correlation analysis shows that older patients have significantly higher maximal 307 pain levels, being contrary to current literature, emphasising young age to be a risk factor 308 for postoperative pain11,25. A higher tolerance of pain is awarded to older patients with 309 a reduced analgetic consumption, a reduced nociceptive activity and lower needs for mor-310 phine medication26. Elderly patients vary in distribution of medication, metabolism and 311 excretion of pain medication compared to young people27.

References are outdated. Please provide more recent literature.

L2656-267 Morphology wise older skin shows dermal atrophy with less perfusion, less elastic fibres and less Meissner- and Vater Pacini bodies resulting in a lower tactile and pressure sensibility27,28.

Do authors mean: resulting in a lower tactile and pressure sensibility tresholds? Otherwise I do not understand.

 We thank the reviewer for this comment. In deed we think that having less sensibility in skin weakens the elastisty and the resistance when skin is brought to tension like in the Abdominoplasty. Due to the lack of Tactile bodies one can assume lower threshold  for sensibility.

It is still confusing. Isn’t it that a lack of tactile bodies leads to less sensibility or higher tresholds for sensibility?

 L274-280 The expectations for this procedure due to their payment is higher and probably the conditions of their pain-ranking are much stricter than those in the insurance paid comparative group. Shorter operation time means decline of narcotic effect and less cumulative analgetic dose than longer duration of surgery. Due to less narcosis probands mobilise themselves earlier and this leads to wound tension with worse pain outcome portrayed by increased claim for painkillers and their mood disturbance. The removal of suction drain itself causes pain, discomfort and anxiety30.

Authors involve various factors in an hypothetic explanation theory that may be questioned. Please comment on this.

 We thank you for this insightful comment. Firstly we wanted to express our day to day experience with expectations of self-paid aesthetic compared to insured abdominoplasty. Secondly we mentioned the effect of a longer operation time and the decline of the analgetic medication.

Unfortunately I don not agree with these suggestions.

The expectations for this procedure due to their payment is higher and probably the conditions of their pain-ranking are much stricter than those in the insurance paid comparative group. Authors suggest here that paying and insurance is correlated to pain intensity. Might be, but try to support this or present this less with less certainty.

Although thoughts may be presented, authors should try to find support for it in literature. With respect to authors reply:  Firstly we wanted to express our day to day experience with expectations, they should state so in he discussion itself, but still try to make a dicussion more than a statement just supported by own experience.

 Shorter operation time means decline of narcotic effect and less cumulative analgetic dose than longer duration of surgery.

In anesthesia at the moment the anaesthesiologist proveds taylor made anaesthesia. And pain management should be adequately time and intensity related for its effect. Because pain measurement is on day one after surgery the effects of narcotics should be negligible.

 Due to less narcosis probands mobilise themselves earlier and this leads to wound tension with worse pain outcome portrayed by increased claim for painkillers and their mood disturbance. The removal of suction drain itself causes pain, discomfort and anxiety.

Please replace probands by subjects as you proposed yourself. And also here less narcosis refers to anaesthesia a day earlier, Therefore this suggestion does not hold as the pain assessment is on day one.

 For all these suggestions counts that a sound discussion in scientific literature needs comments derived from literature. This lacks and makes authors doing unsupported comments.

 Author Response

I thank the authors for their reply. Many issues are addressed, but to my opinion in general the discussion remains of quite poor quality with unsupported suggestions. 

L139-140 In total 7 (13%) of patients stated regularly intake of painkillers before surgery, due to chronic diseases.

Can authors describe if there were patients with chronic pain disorders? If so, which and what treatment did they have before starting the surgery?

We thank the reviewer for the important comment. 13% of patients marked the question E14 of the QUIPS as you may see in the Appendix B “Have you ever had chronic pain before the operation” They only mentioned this but there was no hint for further/extensive medication treatment in this group.

It is a pity that Quips does not provide further information about pre-existing chronic disorders that ask for painkillers, as this might contribute to understanding the postoperative course in these patients.

 We thank the reviewer for this comment and fully agree! Unfortunately the used QUIPS has been designed for postoperative pain assessment- therefore no conclusion can be drawn about pre-existing chronic disorders. This was added as a limitation in the discussion section.

------------------------------

L143 Patient satisfaction is reported to be 11.95 ± 3.03 in average. 11.95?

On what scale is this?

There is the Question E13 in the QUIPS which has a patient satisfaction scale from 0-15

Why is this a scale  from 0-15, while usually these scales run form 0-10?

We thank the reviewer for raising this question. The scale is given by the standardized QUIPS (attached as supplemental data)- therefore we can´t really answer this question.

-------------------------------

S3-guideline is an unknown term, although a reference is provided. Can authors give a more appropriate description?

We thank the reviewer for this important comment. The german medical guideline system contents of 3 levels of evidence. For this reason this S3-Guideline is regarded as a standard in perioperative pain treatment.

I thank the authors for this explanation. However, for the international reader S3 is still unknown. So authors should give a reference or a short explanation in the text.

We thank the reviewer for this comment. A reference and a short description is given in the mentioned section.

------------------------------

The correlation analysis shows that older patients have significantly higher maximal 307 pain levels, being contrary to current literature, emphasising young age to be a risk factor 308 for postoperative pain11,25. A higher tolerance of pain is awarded to older patients with 309 a reduced analgetic consumption, a reduced nociceptive activity and lower needs for mor-310 phine medication26. Elderly patients vary in distribution of medication, metabolism and 311 excretion of pain medication compared to young people27.

References are outdated. Please provide more recent literature.

We thank the reviewer for raising this concern. The references have been updated.

L2656-267 Morphology wise older skin shows dermal atrophy with less perfusion, less elastic fibres and less Meissner- and Vater Pacini bodies resulting in a lower tactile and pressure sensibility27,28.

Do authors mean: resulting in a lower tactile and pressure sensibility tresholds? Otherwise I do not understand.

 We thank the reviewer for this comment. In deed we think that having less sensibility in skin weakens the elastisty and the resistance when skin is brought to tension like in the Abdominoplasty. Due to the lack of Tactile bodies one can assume lower threshold  for sensibility.

It is still confusing. Isn’t it that a lack of tactile bodies leads to less sensibility or higher tresholds for sensibility?

Indeed, we apologize for the confusion. Lack of tactile bodies leads to higher sensibility thresholds and thereby lower sensibility. This is stated in the manuscript.

L274-280 The expectations for this procedure due to their payment is higher and probably the conditions of their pain-ranking are much stricter than those in the insurance paid comparative group. Shorter operation time means decline of narcotic effect and less cumulative analgetic dose than longer duration of surgery. Due to less narcosis probands mobilise themselves earlier and this leads to wound tension with worse pain outcome portrayed by increased claim for painkillers and their mood disturbance. The removal of suction drain itself causes pain, discomfort and anxiety30.

Authors involve various factors in an hypothetic explanation theory that may be questioned. Please comment on this.

 We thank you for this insightful comment. Firstly we wanted to express our day to day experience with expectations of self-paid aesthetic compared to insured abdominoplasty. Secondly we mentioned the effect of a longer operation time and the decline of the analgetic medication.

Unfortunately I don not agree with these suggestions. 

The expectations for this procedure due to their payment is higher and probably the conditions of their pain-ranking are much stricter than those in the insurance paid comparative group. Authors suggest here that paying and insurance is correlated to pain intensity. Might be, but try to support this or present this less with less certainty.

Although thoughts may be presented, authors should try to find support for it in literature. With respect to authors reply:  Firstly we wanted to express our day to day experience with expectations, they should state so in he discussion itself, but still try to make a dicussion more than a statement just supported by own experience.

 We thank the reviewer for this comment. The whole paragraph was restructured.

Shorter operation time means decline of narcotic effect and less cumulative analgetic dose than longer duration of surgery. 

In anesthesia at the moment the anaesthesiologist proveds taylor made anaesthesia. And pain management should be adequately time and intensity related for its effect. Because pain measurement is on day one after surgery the effects of narcotics should be negligible.

 Thank you for this comment. This sentence was removed.

Due to less narcosis probands mobilise themselves earlier and this leads to wound tension with worse pain outcome portrayed by increased claim for painkillers and their mood disturbance. The removal of suction drain itself causes pain, discomfort and anxiety.

Please replace probands by subjects as you proposed yourself. And also here less narcosis refers to anaesthesia a day earlier, Therefore this suggestion does not hold as the pain assessment is on day one.

 We thank the reviewer for raising this concern. We rewrote the section.  

For all these suggestions counts that a sound discussion in scientific literature needs comments derived from literature. This lacks and makes authors doing unsupported comments.

We thank the reviewer for significantly improving our manuscript. All comments have been implemented.